# Effect of Preoperative Anxiety on Postoperative Pain after Craniotomy

**DOI:** 10.3390/jcm11030556

**Published:** 2022-01-22

**Authors:** Lucía Valencia, Ángel Becerra, Nazario Ojeda, Ancor Domínguez, Marcos Prados, Jesús María González-Martín, Aurelio Rodríguez-Pérez

**Affiliations:** 1Department of Anesthesiology, University Hospital of Gran Canaria Doctor Negrín, 35010 Las Palmas de Gran Canaria, Spain; angbecbol@gmail.com (Á.B.); nojebet@gobiernodecanarias.org (N.O.); ancor.dominguez@gmail.com (A.D.); marcospm_9@hotmail.com (M.P.); arodperp@gobiernodecanarias.org (A.R.-P.); 2Department of Medical and Surgical Sciences, University of Las Palmas de Gran Canaria, 35016 Las Palmas de Gran Canaria, Spain; 3Research Unit, University Hospital of Gran Canaria Doctor Negrín, 35010 Las Palmas de Gran Canaria, Spain; josu.estadistica@gmail.com

**Keywords:** pain, anxiety, craniotomy

## Abstract

Pain following craniotomy is challenging. Preoperative anxiety can be one of the controllable factors for prevention of post-craniotomy pain. The main objective of this prospective observational study is to determine this relationship in patients undergoing scheduled craniotomy from February to June 2021. After excluding patients with Mini-Mental State Examination (MMSE) ≤ 24 points, we administered a preoperative State Trait Anxiety Inventory (STAI) questionnaire. We recorded the patient’s analgesic assessment using the Numerical Rating Score (NRS) at 1, 8, 24, and 48 h after surgery. A total of 73 patients were included in the study. Twelve others were excluded due to a MMSE ≤ 24 points. The main predictors for NRS postoperatively at 1, 8, 24, and 48 h were STAI A/E score, male gender, youth, and depression. We identified a cut-off point of 24.5 in STAI A/E for predicting a NRS > 3 (sensitivity 82% and specificity 65%) at 24 h postoperative and a cut-off of 31.5 in STAI A/R (sensitivity 64% and specificity 77%). In conclusion, preoperative STAI scores could be a useful tool for predicting which patient will experience at least moderate pain after craniotomy. The identification of these patients may allow us to highlight psychological preparation and adjuvant analgesia.

## 1. Introduction

Craniotomy has been considered less painful than other surgical interventions because of the lack of nociceptors in the brain tissue. However, we now know that this is not true. The incidence of postoperative pain is high, and its intensity can range from moderate to severe [1]. Recent studies as well as expert opinions confirm that post-craniotomy pain has been under-treated and poorly managed [2,3,4].

Inadequate management of acute postoperative pain is deleterious since it increases morbidity, changes in quality of life, functionality, and delays in recovery time. It is also unfavorable for the healthcare system because it increases costs [5]. Specifically, postoperative pain can have harmful consequences in neurosurgical patients as it may imply the development of postoperative agitation and the elevation of blood pressure levels. This increases the risk of postoperative cerebral hemorrhage [4]. Furthermore, poor acute pain control in these patients increases the risk of chronic headaches, with an incidence of up to 23% to 34% three months after the intervention [1]. Therefore, management of acute pain after craniotomy must be meticulous.

One of the controllable factors for prevention of post-craniotomy pain is preoperative anxiety. A relationship between preoperative anxiety and acute postoperative pain in surgeries other than craniotomies has been defined [6]. Anxiety may be higher in neurosurgery since neurosurgical patients face not only cancer, but also the fear of developing neurological squeals [7,8]. The presence of preoperative anxiety in neurosurgical patients is associated with a poorer quality of life, cognitive performance, memory and attention capacity, longer hospitalization time, depression, and increased physical disability. However, the relationship between preoperative anxiety and the development of postoperative pain has not been studied sufficiently [9].

The main objective of this prospective observational study is to determine the relationship between preoperative anxiety and postoperative pain in patients undergoing scheduled craniotomy under general anesthesia. As secondary objectives, we assess preoperative risk factors for preoperative anxiety, and which perioperative variables can predispose a patient to suffer from greater postoperative pain.

## 2. Materials and Methods

This prospective observational study evaluated preoperative anxiety and postoperative pain in patients scheduled to undergo craniotomy in a university hospital from February to June 2021. The study was approved by the Ethics Committee of Hospital Universitario de Gran Canaria Doctor Negrín, Las Palmas de Gran Canaria, Spain (approval #2019-241-1, Chairperson Dr. Fiuza), and prospectively registered at Clinicaltrials.gov, accessed on 22 December 2021 (NCT04720248). All methods followed good clinical practice. All patients over 18 years old who were to undergo scheduled supratentorial craniotomy and who signed informed consent were included. Exclusion criteria were: patients suffering from disabilities or cognitive impairment defined as a score ≤24 points in the Mini-Mental State Examination (MMSE), or patients who could not collaborate on the postoperative clinical assessment. This manuscript follows the STROBE guidelines [10].

### 2.1. Outcomes

Preoperative anxiety was evaluated using the Anxiety Scale State-Trait Anxiety Inventory (STAI), a questionnaire developed by Spielberg et al. in 1970 [11]. A Spanish version of STAI has been validated for the Spanish population [12]. This questionnaire is considered an instrument to study anxiety through a self-evaluation of two independent concepts: State Anxiety and Trait Anxiety. State Anxiety (STAI A/E) is conceptualized as a transitory emotional condition characterized by tension, apprehension, and hyperarousal of the Autonomous Nervous System. It can vary in intensity and fluctuate over time. Trait Anxiety (STAI A/R) is characterized by a stable anxious propensity due to the subject’s tendency to perceive everyday situations as threatening, thus causing an increase in the degree of anxiety. As anxiety is a subjective quality dependent on age and gender, it has been established that a STAI score of 20 corresponds to the 50th percentile in the adult population [13]. Postoperative pain was assessed per protocol by the Acute Pain Unit using the Numerical Rating Score (NRS), measuring pain from 0 (no pain) to 10 (maximum pain imaginable).

### 2.2. Study Protocol

Upon hospital admission, an independent investigator performed the MMSE. If the score in this examination was higher than or equal to 24 points, preoperative anxiety was evaluated using the STAI questionnaire. After administering the STAI questionnaire, patients were instructed on how to perform postoperative NRS to evaluate postoperative pain. Demographic preoperative data were collected: age, gender, weight, height, body mass index (BMI), educational level (dichotomized as elementary education or higher education), and physical status score according to the American Society of Anesthesiologists (ASA) (from 1 to 5). The patient’s medical history was also recorded: depression, anxiety, rheumatological illnesses, and previous use of drugs such as metamizole, corticosteroids, antidepressants, and chronic use of analgesics. Histopathological results of the craniotomy were also collected.

All patients underwent elective craniotomy under general anesthesia. The diagnostic (vascular or tumoral) and the type of craniotomy was recorded according to the surgical incision (whether it was frontal/temporal or parietal/occipital), as well as the duration of the intervention. Intraoperative anesthetic management was carried out according to standard clinical practice. Patients were monitored with electrocardiogram, peripheral oxygen saturation, invasive arterial pressure, and bispectral index (BIS, Covidien, Dublin, Ireland). General anesthesia was performed using continuous propofol infusion to maintain BIS 40–60, using cisatracurium in bolus (0.2 mg·kg^−1^) to allow orotracheal intubation and in continuous infusion during surgery to ensure intraoperative immobility. As analgesics, continuous infusion of remifentanil or boluses of fentanyl or tramadol were used following clinical practice to maintain intraoperative hemodynamic stability. Patients also received intraoperative non-opioid analgesic drugs (metamizole or paracetamol). If corticosteroids were given intraoperatively, it was also recorded along with the dosage. At the end of the surgery, the neuromuscular blockade was reversed and the tracheal tube was removed. The appearance of intraoperative complications and the placement or not of drains were recorded. The anesthesiologist in charge of the intraoperative management of the patient was not aware that the patient was participating in a study evaluating postoperative analgesia. Patients were transferred to the postoperative Intensive Care Unit (ICU) under the care of an independent clinician. They were transferred to the surgical ward 24 h after the surgery. In the absence of allergies, all patients received an endovenous continuous infusion of metamizole 12 g during the first 48 postoperative hours. If patients were allergic to metamizole, paracetamol 1g each 6 h was prescribed. An investigator, who was blinded to the patient’s preoperative anxiety assessment, collected the patient’s analgesic assessment using NRS at 1, 8, 24, and 48 h after surgery. Follow-up ended 48 h postoperatively.

### 2.3. Statistical Analysis

Data were analyzed using the statistical program R Core Team 2021, version 4.1.0 Vienna, Austria. Data on quantitative variables are expressed as mean, standard deviation, median, and 25–75th percentiles. The Kolmogorov test was used to verify the normality of the data. Qualitative variables are described as absolute and relative frequencies. The T-Student test was used to compare quantitative variables between two groups, and Fisher’s exact test was used to check the relationship between qualitative variables. Multiple linear regression was used to predict quantitative variables. To perform the predictive model for the NRS variable at 1, 8, 24, and 48 h after surgery, the following variables were analyzed: STAI A/E, STAI A/R, age, female gender, tumoral diagnostic, ASA physical status 3–4, higher educational level, suffering from rheumatological illnesses, depression, or anxiety, previous use of metamizole and corticosteroids, BMI, MMSE, parietal/occipital craniotomy, length of surgery, placement of drains at the end of surgery, and the use of intraoperative corticoids dichotomized into 4 mg or more than 4 mg. The forward-backward technique was used to select the optimal model. To check the multicollinearity of the variables, VIF statistic was used, and it was valid if this was lower than 7. The area under the curve (AUC) was calculated to find the best cut-off values of the STAI A/E and STAI A/R to predict the NRS ≥3 at the first and 24 h postoperatively and expressed as AUC. Sensitivity and specificity were calculated to choose the best cut-off values. A *p*-value < 0.05 was considered statistically significant.

## 3. Results

From the 85 patients assessed for eligibility during the study period, 12 were excluded due to a score of ≤24 points in the MMSE. Consequently, 73 patients were included in the study. The characteristics of the patients and the intraoperative variables are summarized in Table 1. No statistically significant differences were found between the preoperative characteristics of patients regarding the STAI A/E and STAI A/R values dichotomized into less than 20 or more than or equal to 20 (Table 2 and Table 3).

The incidence of nausea was 1.4%. No postoperative vomiting was recorded. Pain assessment was performed postoperatively on all patients. The mean NRS was 3.26 + 1.97 at the first hour, 2.56 + 1.71 at 8 h, 2.04 + 1.46 at 24 h, and 1.84 + 1.20 at 48 h. The optimal variables to predict postoperative NRS were calculated. The optimal variables found were: STAI A/E, age, female gender, ASA physical status 3–4, depression, and MMSE for NRS at the first postoperative hour; STAI A/E, age, female gender, depression, parietal/occipital craniotomy, and length of surgery for the NRS at 8 postoperative hours; STAI A/E, age, female gender, ASA physical status 3–4, depression, parietal/occipital craniotomy, and length of surgery for the NRS at 24 postoperative hours; and STAI A/E, age, length of surgery, and use of intraoperative dexamethasone for NRS at 48 postoperative hours. Table 4 and Table 5 show the coefficients of the univariate and multivariate analyses of the variables that influence the different NRS evaluated in the postoperative period.

We identified that a cut-off point of 24.5 in STAI A/E was the best one to predict NRS ≥ 3, with an area under the curve (AUC) of 0.60 with sensitivity 62% and specificity 65% at the first postoperative hour, and an AUC of 0.69 with a sensitivity 82% and specificity 65% at the assessment performed at 24 postoperative hours. However, in the STAI A/R questionnaire, 22.5 was the cut-off point that best defined a NRS greater than 3 in the first postoperative hour, with an AUC of 0.5 and with a sensitivity 33% and a specificity 77%, and the cut-off point of 31.5 was better to define NRS at 24 postoperative hours with an AUC of 0.68, sensitivity 64%, and specificity of 77% (Figure 1 and Figure 2, ROC curves).

## 4. Discussion

In this prospective observational study, we demonstrated that a STAI A/E score greater than 24.5 predicts which patients will experience at least moderate pain after craniotomy. In addition, we found that the presence of depression, youth, and male gender can worsen postoperative pain. No relationship was found between preoperative variables and anxiety levels.

Postoperative headache is a common complication after surgical procedures other than craniotomy. Several risk factors have been demonstrated to be related to this complication, such as age, female gender, previous history of headache, and anesthetic drugs. On the other hand, post-craniotomy pain is a different and more specific entity. Its management is challenging for anesthesiologists. One of the main reasons for poor therapeutic control is the fear of the side effects of analgesics drugs. The use of opioids can interfere with neurological examination, andnon-steroidal anti-inflammatories (NSAIDs) can increase the risk of bleeding, which is highly dangerous in neurosurgical patients. Given that the pharmacological management of pain control is a challenge, several studies have focused on discovering other variables that could intensify or mitigate pain. Looking for strategies to improve pain control, ERAS (Enhanced Recovery after Surgery) programs have also reached neurosurgery [14]. It has recently been demonstrated that patients included in ERAS programs show, among many other advantages, less postoperative pain [9]. It is interesting to note that the literature reports that acute pain after craniotomy is moderate-severe. However, opioids can be reduced or avoided in ERAS protocols, resulting in an even better postoperative pain control. In the ERAS program, patient education to adjust perioperative expectations has been shown to increase patient satisfaction [15]. This intervention might be the reason preoperative anxiety and, consequently, postoperative pain are mitigated. In our study, we have been able to identify those patients who will experience more postoperative pain by using a preoperative anxiety test. These patients could benefit from several actions in order to optimize postoperative control.

On one side, the use of adjunctive therapies could help ameliorate pain. In a recent systematic review of pharmacological interventions for the prevention of acute postoperative pain in adults following craniotomy, it has been indicatedthat NSAIDs rank first for their efficacy [1]. However, there are alternatives that are gaining strength, and that can reduce postoperative analgesic requirements. Intraoperative use of dexmedetomidine has been demonstrated to improve analgesia management, reducing opioid consumption during the first 24 h after craniotomy [16,17]. Another multimodal analgesic is the scalp block. Although infiltration of the surgical wound with local anesthetics may be easier to perform and have similar efficacy, the analgesic effect offered by scalp block is superior and longer [18].

On the other hand, as we have already mentioned, psychological preparation prior to surgery can reduce postoperative pain due to its psychological component. This may involve several strategies described in previous studies [19]: providing information on the procedures and equipment to be used, talking about care expectations and past stressful experiences, reporting how pain is managed postoperatively with the description of analgesic techniques, and encouraging requesting painkillers. There is a Cochrane systematic review confirming that the use of techniques focused onproviding better psychological preparation areassociated with less postoperative pain, but with low evidence [19]. However, the same authors recommend caution upon interpreting the results. This is due to the heterogeneity of the psychological preparations and outcomes of the articles included in the review, as most were carried out for cardiac and orthopedic surgery. In the field of neurosurgery, there is a study on the psychological consequences of awake craniotomy. This prospective study found a positive correlation between preoperative anxiety according to the Hospital Anxiety and Depression Scale (HADS) and the NRS on the third postoperative day [20]. However, it is striking that, in this study, pain assessment was carried out at the third postoperative day, considering that the highest incidence of pain occurs on the first postoperative day, as we demonstrated in our study. The scarcity of studies in neurosurgery might be due to the difficulty ofachieving a sufficient sample size because neurosurgical procedures account for only 2.6% of all surgeries [21]. Furthermore, cognitive impairment is highly prevalent in patients undergoing craniotomy [22] as a consequence of the presence of tumor, tumor-related epilepsy, use of corticosteroids, or increased intracranial pressure [23]. Cognitive impairment makes pain and anxiety scales difficult to perform. Thus, in our work, patients with MMSE ≤ 24 points were excluded.

Regarding the scale chosen to assess anxiety, most prospective studies on preoperative anxiety have used STAI. [11]. However, there are other scales to assess preoperative anxiety. The “Amsterdam preoperative anxiety and information scale” (APAIS) [24] is a scale specifically designed to detect anxiety related to anesthesia and surgery using 6 simple questions. Another scale is HADS [25], which is the international gold standard for assessing anxiety and depression in patients suffering from physical complaints. Both scales, APAIS and HADS, could be reflected into the STAI A/E and STAI A/R questionnaires, respectively; while the APAIS would be the equivalent of the STAI A/E, the HADS would be similar to STAI A/R, since it reflects how the individual feels in a general way. Therefore, the STAI scale is the most adequate since both anxiety as a temporary condition experienced in a specific situation (STAI A/E) as well as anxiety as the generalized tendency to perceive situations as threatening (STAI A/R) can play a role in experiencing postoperative pain [26,27]. One study assessed the quality of different instruments to assess preoperative anxiety in neurosurgical patients including STAI, APAIS, or the assessment of anxiety on a numerical scale (NRS), concluding that all are valid but that the STAI questionnaire would be the ideal instrument for evaluation ifcase reliability was of particular importance [28].

In this study, we observed that both STAI A/E and STAI A/R are capable of predicting that the patient will experience at least moderate pain at 24 h, with an 82% and 64% sensitivity and a 65% and 77% specificity, respectively. Given these sensitivity and specificity values, if one of the two had to be chosen, we would select the most sensitive (STA A/E). In addition, the cut-off point of the STAI A/R is too high to be used, so few patients would obtain this score and would benefit from this screening.

Regarding the variables that could affect post-craniotomy pain, we found that the most important was the STAI A/E score. However, we also stated that other factors can influence postoperative pain such as male gender, youth, and depression. Young patients have a higher incidence of pain. Currently, it is known that the probability of experiencing post-craniotomy pain is reduced by 3% for each year of life [29]. It has been also shown that the presence of anxiety, depression, and postoperative pain can be risk factors for the worsening of post-craniotomy pain [4]. In our study, unlike in other studies, men experienced more pain after craniotomy. No explanation for this finding was found.

Analyzing the risk factors for preoperative anxiety, we did not find a significant relationship with the preoperative variables analyzed. In many studies, women have higher levels of anxiety than men [9,28]. We also found a higher percentage of women in the group with the highest STAI, although these differences were not significant. Wedid not find a relationship between preoperative anxiety and histological diagnosis, as in previous articles [9].

We acknowledge some potential limitations in the present study. First, the incidence of chronic pain 3 months postoperatively was not analyzed. Considering the high incidence of chronic pain after craniotomy, this outcome could have been interesting and relevant, since its presence actually reduces the quality of life of our patients. Second, excluding patients using the MMSE may be a quick technique, but could also be inaccurate. It has been reported that cognitive impairment is frequent in patients who are going to undergo craniotomy [23] and for this reason, in our study, 14% of screened patients were excluded. The use of more comprehensive tests could have excluded more patients with more specific results. However, cognitive deficits secondary to psychological disturbances are also described. Taking into account the relationship that anxiety may have with other psychological disorders, it may have produced a bias in the exclusion of patients with cognitive deficits related to the main variable under study.Third, although the STAI is the most appropriate test to carry out this study, we are aware that the duration of the test (approximately 20 min) is rather long. This could lead to inaccurate responses due to patients becoming tired. In addition, this test may be too long for use in routine clinical practice.

## 5. Conclusions

In conclusion, this is the first study in patients scheduled for craniotomy under general anesthesia showing that preoperative STAI scores could be a useful tool for predicting which patient will experience at least moderate pain after craniotomy. The early identification of these patients may allow us to insist on several aspects, such as psychological preparation and adjuvant analgesia. A preoperative visit performed by a specialist (nurse, neurosurgeon, or anesthesiologist) with a psychologist could reduce preoperative anxiety and, thus, influence the appearance of postoperative pain.

## Figures and Tables

**Figure 1 jcm-11-00556-f001:**
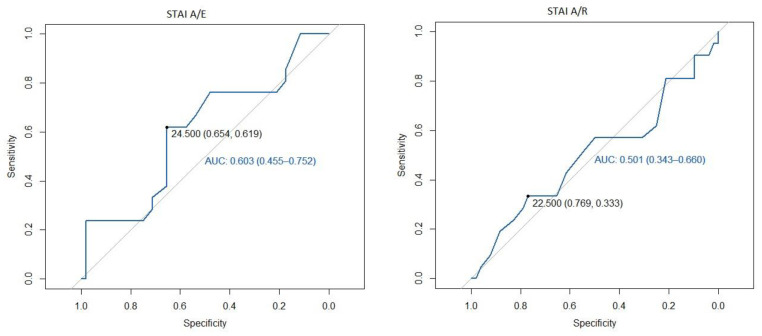
ROC curve of the STAI at 1 h postoperative as a predictor of mild pain. STAI A/E: State-Trait Anxiety Inventory State Anxiety; STAI A/R: State-Trait Anxiety Inventory Trait Anxiety; AUC: Area under the curve.

**Figure 2 jcm-11-00556-f002:**
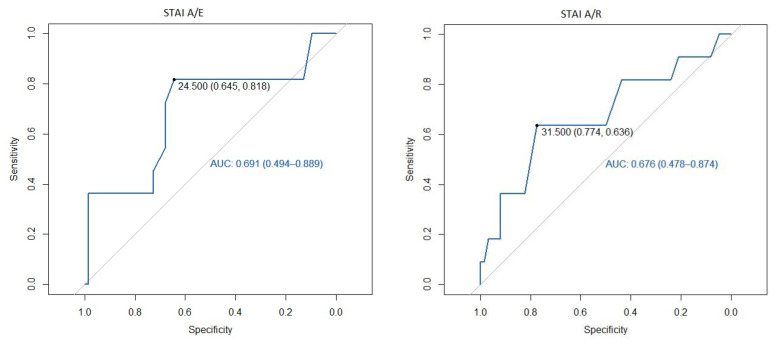
ROC curve of the STAI at 24 h postoperative as a predictor of mild pain. STAI A/E: State-Trait Anxiety Inventory State Anxiety; STAI A/R: State-Trait Anxiety Inventory Trait Anxiety.

**Table 1 jcm-11-00556-t001:** Characteristics of patients and intraoperative variables.

		Study Population (*n* = 73)
Age, years		55 ± 15
Female, No. (%)		45 (61.6)
BMI, kg·m^−1^		24.9 ± 5.5
ASA 3, No. (%)		38 (52.1)
Medical history		
	Depression, No. (%)	16 (21.9)
	Anxiety, No. (%)	5 (6.9)
	Rheumatological diseases, No. (%)	4 (5.5)
Previous use of drugs		
	Metamizole, No. (%)	27 (36.9)
	Corticosteroids, No. (%)	22 (30.1)
	Antidepressants, No. (%)	14 (19.2)
	Chronic use of analgesics, No. (%)	12 (16.4)
Educational level: Basic elementary education, No. (%)		43 (58.9)
MMSE, points		28.9 ± 1.3
STAI A/E, score		22.4 ± 10.8
STAI A/R, score		26.9 ± 7.5
Main diagnosis: tumoral, No. (%)		68 (93.2)
Craniotomy frontal/temporal, No. (%)		51 (69.9)
Length of surgery, min		241 ± 89
Intraoperative drugs		
	Remifentanil 0.1 mcg·kg^−1^·min^−1^, No. (%)	27 (37.0)
	Remifentanil 0.1 mcg·kg^−1^·min^−1^ + Fentanyl, No. (%)	42 (57.5)
	Remifentanil 0.1 mcg·kg^−1^·min^−1^ + Tramadol, No. (%)	4 (5.5)
	Not remifentanil, No. (%)	19 (26.0)
	Dexamethasone 4 mg, No. (%)	29 (39.7)
	Dexamethasone >4 mg, No. (%)	25 (34.3)
Postoperative drain (%)		45 (61.6)

Data are expressed as mean + SD, or frequency (%). BMI: body mass index; ASA: American Society of Anesthesiologists; MMSE: Mini-Mental State Examination; STAI A/E: State-Trait Anxiety Inventory State Anxiety; STAI A/R: State-Trait Anxiety Inventory Trait Anxiety.

**Table 2 jcm-11-00556-t002:** Comparison of preoperative characteristics of patients between STAI A/E <20 or >20.

Variable		STAI A/E < 20 (*N* = 29)	STAI A/E ≥ 20 (*N* = 44)	*p*-Value
Age, years		58 ± 15	54 ± 15	0.250
Gender				
	Female, No. (%)	16 (55.2)	29 (65.9)	0.462
	Male, No. (%)	13 (44.8)	15 (34.1)	
BMI, kg·m^−1^		24.6 ± 4.7	25.0 ± 6.1	0.767
Educational level				
	Basic elementary, No. (%)	20 (69.0)	23 (52.3)	0.224
	Other No. (%)	9 (31.0)	21 (47.7)	
MMSE, points		29.0 ± 1.1	28.9 ± 1.5	0.886
Medicalhistory				
	Depression, No. (%)	5 (17.2)	11 (25)	0.567
	Anxiety, No. (%)	2 (6.9)	3 (6.8)	1.000
	Rheumatological diseases, No. (%)	2 (6.9)	2 (4.5)	0.66
Previous useof drugs				
	Metamizole, No. (%)	8 (27.6)	19 (43.2)	0.220
	Corticosteroids, No. (%)	6 (20.7)	16 (36.4)	0.197
	Antidepressants, No. (%)	6 (20.7)	8 (18.2)	1.000
	Chronic use of analgesics, No. (%)	7 (24.1)	5 (11.4)	0.200

Data are expressed as mean + SD, or frequency (%). STAI A/E: State-Trait Anxiety Inventory State Anxiety; BMI: body mass index; MMSE: Mini-Mental State Examination.

**Table 3 jcm-11-00556-t003:** Comparison of preoperative characteristics of patients between STAI A/R <20 or >20.

Variable		STAI A/R < 20 (*N* = 10)	STAI A/R ≥ 20 (*N* = 63)	*p*-Value
Age, years		53 ± 18	56 ± 14	0.659
Gender				
	Female, No. (%)	5 (50)	23 (36.5)	0.492
	Male, No. (%)	5 (50)	40 (63.5)	
BMI, kg·m^−1^		25.7 ± 4.3	24.7 ± 5.7	0.480
Educational level				
	Basic elementary education, No. (%)	7 (70)	36 (57.1)	0.510
	Other, No. (%)	3 (30)	27 (42.9)	
MMSE, points		29.1 ± 0.9	28.9 ± 1.4	0.885
Medicalhistory				
	Depression, No. (%)	1 (10)	15 (23.8)	0.443
	Anxiety, No. (%)	1 (10)	4 (6.3)	0.532
	Rheumatological diseases, No. (%)	1 (10)	3 (4.8)	0.499
Previous use of drugs				
	Metamizole, No. (%)	4 (40)	23 (36.5)	1.000
	Corticosteroids, No. (%)	1 (10)	21 (33.3)	0.264
	Antidepressants, No. (%)	2 (20)	12 (19)	1.000
	Chronic use of analgesics, No. (%)	2 (20)	10 (15.9)	0.665

Data are expressed as mean + SD, or frequency (%). STAI A/R: State-Trait Anxiety Inventory Trait Anxiety; BMI: body mass index; MMSE: Mini-Mental State Examination.

**Table 4 jcm-11-00556-t004:** Main predictors for NRS 1 and 4 h postoperatively.

	NRS at 1 h				NRS at 8 h			
Variable	Univariate	*p*-Value	Multivariate	*p*-Value	Univariate	*p*-Value	Multivariate	*p*-Value
STAI A/E	0.08 (0.04–0.11)	<0.001	0.06 (0.02–0.1)	0.003	0.04 (0.01–0.08)	0.024	0.04 (0–0.08)	0.029
STAI A/R	0.05 (−0.01–0.11)	0.09	−0.01 (−0.07–0.05)	0.76	0.04 00280.01–0.09)	0.13	0.03 00280.06–0.06)	0.968
Age	–0.06 (−0.09–−0.03)	<0.001	−0.07 (−0.1–−0.04)	<0.001	−0.04 (−0.07–−0.02)	0.001	−0.05 (−0.08–−0.03)	<0.001
Female gender	0.07 (−0.88–1.02)	0.88	−0.7 (−1.55–0.15)	0.11	−0.19 (−1.01–0.63)	0.648	−0.72 (−1.5–0.06)	0.068
Depression	0.23 (–0.89–1.34)	0.69	0.93 (−0.05–1.9)	0.06	0.8 (–0.15–1.75)	0.097	1.48 (0.6–2.36)	0.001
ASA physical status 3–4	0.32 (−0.6–1.24)	0.49	0.63 (−0.13–1.4)	0.1	-	-	-	-
MMSE	–0.03 (−0.38–0.32)	0.87	–0.25 (−0.56–0.06)	0.11	-	-	-	-
Parietal/occipital craniotomy	-	-	-	-	0.04 (−0.83–0.92)	0.924	0.76 (−0.02–1.54)	0.056
Length of surgery	-	-	-	-	0 (−0.01–0)	0.138	−0.003 (−0.01–0)	0.163

Data are expressed as coefficient (95% CI). NRS: Numerical Rating Score; STAI A/E: State-Trait Anxiety Inventory State Anxiety; STAI A/R: State-Trait Anxiety Inventory Trait Anxiety; MMSE: Mini-Mental State Examination; ASA: American Society of Anesthesiologists.

**Table 5 jcm-11-00556-t005:** Main predictors for NRS 24 and 48 h postoperatively.

	NRS at 24 h				NRS at 48 h			
Variable	Univariate	*p*-Value	Multivariate	*p*-Value	Univariate	*p*-Value	Multivariate	*p*-Value
STAI A/E	0.04 (0.01–0.07)	0.018	0.04 (0–0.07)	0.029	0.03 (0–0.05)	0.053	0.02 (0–0.05)	0.078
STAI A/R	0.04 (−0.01–0.08)	0.083	0.02 (−0.03–0.07)	0.502	0.04 (0–0.07)	0.061	0.01 (−0.03–0.06)	0.53
Age	−0.04 (0.06–−0.02)	<0.001	−0.04 (−0.06–−0.02)	<0.001	−0.02 (−0.04–0)	0.069	−0.01 (−0.03–0)	0.145
Female gender	−0.11 (−0.81–0.6)	0.762	−0.48 (−1.17–0.22)	0.17	-	-	-	-
Depression	0.11 (−0.72–0.93)	0.796	0.61 (−0.17–1.39)	0.12	-	-	-	-
ASA physical status 3–4	0.35 (−0.33–1.03)	0.306	0.44 (−0.17–1.06)	0.153	-	-	-	-
Parietal/occipital craniotomy	0.01 (−0.74–0.75)	0.987	0.54 (−0.15–1.24)	0.123	-	-	-	-
Length of surgery	0 (−0.01–0)	0.02	−0.002 (−0.01–0)	0.183	0 (−0.01–0)	0.009	−0.003 (−0.01–0)	0.025
Intraoperative dexamethasone 4 mg	-	-	-	-	−0.83 (−1.52–−0.14)	0.019	−0.76 (−1.42–−0.1)	0.025
Intraoperative dexamethasone >4 mg	-	-	-	-	−0.44 (−1.15–0.28)	0.226	−0.39 (−1.06–0.29)	0.255

Data are expressed as coefficient (95% CI). NRS: Numerical Rating Score; STAI A/E: State-Trait Anxiety Inventory State Anxiety; STAI A/R: State-Trait Anxiety Inventory Trait Anxiety; ASA: American Society of Anesthesiologists.

## Data Availability

The data that support the findings of this study are available upon request to the corresponding authors.

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
