# Peer review of "Effect of Preoperative Anxiety on Postoperative Pain after Craniotomy"

_jcm, 2022, doi:10.3390/jcm11030556_

Round 1

Reviewer 1 Report

This is a well designed prospective observational study that contributes to the field of post-craniotomy pain research. It showed that preoperative anxiety, depression, young age and male are associated with postoperative pain following craniotomy. The study filled in some knowledge gap between the preoperative psychological condition and the postoperative pain after craniotomy.

The whole manuscript follows pretty well--good job!

I would recommend the authors to make brief comment on the difference between postoperative headache and post-craniotomy pain.

Other minor comments:

  1. P203:" The use of opioids can interfere with neurological monitoring...", do you mean "neurological examination"? if not, how the use of opioids can interfere with neurological monitoring?
  2. P227: "the analgesic effect offered by regional block is superior and longer". Is the regional block referring to scalp block or other blocks?
  3. I would briefly mention the  limitation of using  STAI A/E and STAI A/R in the study.

Author Response

Response to Reviewers: Manuscript ID:  jcm-1545430

Reviewer #1

Q1) This is a well designed prospective observational study that contributes to the field of post-craniotomy pain research. It showed that preoperative anxiety, depression, young age and male are associated with postoperative pain following craniotomy. The study filled in some knowledge gap between the preoperative psychological condition and the postoperative pain after craniotomy. The whole manuscript follows pretty well--good job!

 R1- We truly thanks the Reviewer for the positive comments about our study.

Q2) I would recommend the authors to make brief comment on the difference between postoperative headache and post-craniotomy pain.

R2- Thank you for this comment. As suggested by the Reviewer, it is important to remark the difference between postoperative headache and post-craniotomy pain

 We have added this information in the discussion section.

Q3) Other minor comments: P203:" The use of opioids can interfere with neurological monitoring...", do you mean "neurological examination"? if not, how the use of opioids can interfere with neurological monitoring?

R3- Thanks for this comment. We would like to confirm that we meant “neurological examination”. We have changed this properly in the revised manuscript.

Q4) P227: "the analgesic effect offered by regional block is superior and longer". Is the regional block referring to scalp block or other blocks?

R4- Thank you. We have added scalp instead of regional in the revised manuscript.

Q5) I would briefly mention the  limitation of using  STAI A/E and STAI A/R in the study.

R5- We have added this information as a study limitation in the discussion of the revised manuscript.

Reviewer 2 Report

I have some methodological issues to point. 

The first one is why you did not use scalp block for craniotomy as standard practice in neuranesthesia protocols (in asleep and also awake protocols)

Tramadol is not a good choice for intraoperative analgesia.

What was the incidence of PONV.

Did the neurosurgical pathology differ (oncology and vascular pathology) and what were the differences in the drug consumption.

Best regards

Author Response

Reviewer #2

Q1) I have some methodological issues to point. The first one is why you did not use scalp block for craniotomy as standard practice in neuranesthesia protocols (in asleep and also awake protocols)

R1- Thank you very much for this comment. As this observational study reflects the usual clinical practice in our center, we decided not to modify the clinical practice of the anesthesiologists in charge of the intraoperative management of the patients. As noticed, scalp block had not been incorporated into standard practice in our center in the moment the study was carried out, maybe because our anesthesiologists had not realized its importance until we began to perform awake craniotomies. We have remarked the observational nature of the study in the limitation section of the revised manuscript.

Q2) Tramadol is not a good choice for intraoperative analgesia.

R2- We completely agree with the Reviewer that tramadol is not a good choice for intraoperative analgesia. Its use in the patients included in the study might have been due to our decision not to interfere with the usual clinical practice of the anesthesiologists in charge of the intraoperative management of the patients. However, as it can be observed in the table 1, only 4 patients from the 73 patients included received tramadol as intraoperative analgesia.

Q3) What was the incidence of PONV.

R3- We thank the Reviewer for this comment. The incidence of postoperative nausea observed was 1.4 %. We have now added this information in the Results section of the revised manuscript.

Q4) Did the neurosurgical pathology differ (oncology and vascular pathology) and what were the differences in the drug consumption.

R4- Thank you for this comment. We did not find any difference in NRS in relation to the neurosurgical pathology that led to surgery. I addition, we included this variable (tumoral versus vascular) in the predictive model for the NRS variable at 1, 8, 24 and 48 hours after surgery, resulting not an optimal variable.